# Roles of Progesterone, Testosterone and Their Nuclear Receptors in Central Nervous System Myelination and Remyelination

**DOI:** 10.3390/ijms21093163

**Published:** 2020-04-30

**Authors:** Abdel Mouman Ghoumari, Charly Abi Ghanem, Narimène Asbelaoui, Michael Schumacher, Rashad Hussain

**Affiliations:** U1195 Inserm and University Paris Sud/Paris-Saclay, 80 rue du Général Leclerc, 94276 Kremlin-Bicêtre, France; abighanemc@gmail.com (C.A.G.); narimene.asbelaoui@inserm.fr (N.A.); michael.schumacher@inserm.fr (M.S.); hussainrashad99@yahoo.com (R.H.)

**Keywords:** oligodendrocytes, myelin, multiple sclerosis, steroids, progesterone and testosterone

## Abstract

Progesterone and testosterone, beyond their roles as sex hormones, are neuroactive steroids, playing crucial regulatory functions within the nervous system. Among these, neuroprotection and myelin regeneration are important ones. The present review aims to discuss the stimulatory effects of progesterone and testosterone on the process of myelination and remyelination. These effects have been demonstrated in vitro (i.e., organotypic cultures) and in vivo (cuprizone- or lysolecithin-induced demyelination and experimental autoimmune encephalomyelitis (EAE)). Both steroids stimulate myelin formation and regeneration by acting through their respective intracellular receptors: progesterone receptors (PR) and androgen receptors (AR). Activation of these receptors results in multiple events involving direct transcription and translation, regulating general homeostasis, cell proliferation, differentiation, growth and myelination. It also ameliorates immune response as seen in the EAE model, resulting in a significant decrease in inflammation leading to a fast recovery. Although natural progesterone and testosterone have a therapeutic potential, their synthetic derivatives—the 19-norprogesterone (nestorone) and 7α-methyl-nortestosterone (MENT), already used as hormonal contraception or in postmenopausal hormone replacement therapies, may offer enhanced benefits for myelin repair. We summarize here a recent advancement in the field of myelin biology, to treat demyelinating disorders using the natural as well as synthetic analogs of progesterone and testosterone.

## 1. Introduction

### 1.1. Myelinating Cells of the Central Nervous System

Morphologically, the central nervous system (CNS) can be distinguished into two major regions: grey and white matter. Neuronal cell bodies and dendrites are concentrated in the gray matter, where information is received, processed and integrated, while white matter consists of bundles of axons insulated by a multi-layer lipid structure called myelin. Myelin in CNS is generated by a special type of glia called oligodendrocytes, which extend many process, each of which contacts and repeatedly envelops axons [1]. Thus, as per the requirement of their role, oligodendrocytes are more distributed along the axonal tracks as compared to grey matter regions. Oligodendrocytes can be categorized into two major types: oligodendrocyte progenitor cells (OPCs) and mature myelinating ones. Each category is morphologically distinct; OPCs are bipolar, migratory in nature and proliferative cells while mature oligodendrocytes are highly branched cells specialized to make myelin sheets. They lack the capacity for proliferation and migration. Similarly, mature oligodendrocytes are more prone to damages as compared to OPCs. Importantly, they have limited potential of regeneration in response to any damage to their processes or myelin sheath [2,3,4]. Immuno-histologically, OPCs can be identified by the expression of membranous proteins, platelet-derived growth factor receptor alpha (PDGFRα), A2B5, proteoglycan sulfate (NG2) as well as transcription factor Olig1/2, sox2, Nkx2.2. Mature oligodendrocytes specifically express myelin proteins; proteolipid protein (PLP), myelin basic protein (MBP), myelin oligodendrocyte glycoprotein (MOG), myelin-associated glycoprotein (MAG), adenomatous polyposis coli (APC/CC1), 2′,3′-Cyclic-nucleotide 3′-phosphodiesterase (CNPase) and transcription factor sox10. Several intermediate stages have also been identified during cell transitioning from OPCs to mature and simultaneously expressing multiple proteins that are characteristic of both cell types, and also the morphological characters resembling more the mature oligodendrocytes; one of these intermediate types is termed as pre-oligodendrocytes [2,3,4,5,6,7,8].

During development, OPCs first appear around embryonic day 12.5 in mice, almost the same time in the ventral region of spinal cord named as pMN domain and as well as ventral brain region median ganglionic eminence. The early OPCs proliferate, migrate and preferentially populate the future white matter regions of the CNS. Later on, a second wave from the dorsal regions in the spinal cord and lateral ganglionic eminence in the brain replace or facilitate the whole spread [9,10,11]. Interestingly, these OPCs are capable of symmetric as well as asymmetric division early in development, and the same trend continues in the adult life but much slower and only in response to selective cues arising during injury and rejuvenation [12]. A rough estimate shows that an adult brain consists of 5–10% of these OPCs distributed throughout the brain [13,14]. This density does not perceptibly vary throughout adulthood and remains significantly higher in white matter regions [15]. It is interesting to note that gray matter also contains myelinated axons, very important for proper brain functioning [16]. Indeed, refined MRI techniques have revealed that the severity of disease symptoms in multiple sclerosis (MS) patients is linked to gray matter myelin loss [17]. It could be attributed to the fact that OPCs in white matter are more proliferative than ones in the grey matter [15].

Transitioning from OPCs to mature oligodendrocyte is mainly a later embryonic or early postnatal event; the process is complete by P60 in mice and by adulthood in humans. During the differentiation process, OPCs undergo a morphological transformation, increasing their ramifications or branching, and sequentially expressing different cellular markers and transcription factors. The different developmental stages of the oligodendrocyte lineage have been well characterized based on the following expression of lipid and protein antigens [3,4,18]. In vitro studies showed that mature oligodendrocytes are able to form myelin like membranes and myelin sheaths that enwrap axons as they do in vivo [4,18,19]

Even if the oligodendrocyte plays the main role in the production of myelin, its function is largely influenced by the other glial cells, namely astrocytes and microglia. Astrocytes form stellate cells with multiple processes and occupy about 25% to 50% of brain volume. White matter astrocytes are called fibrous astrocytes, while astrocytes in gray matter are called protoplasmic astrocytes. Besides providing lipids for the synthesis of myelin sheaths during development, astrocytes contribute to the development of oligodendrocytes as well as to myelination and remyelination processes. Intercellular connections between astrocytes and oligodendrocytes are vital for the proper functioning of oligodendrocytes. They produce several growth factors, such as platelet-derived growth factor, brain-derived neurotrophic factor or ciliary neurotrophic factor to promote OPC development and CNS myelination and they aid in the removal of myelin debris [20,21,22,23,24,25]. Astrocytes contribute to the formation and functioning of the blood–brain barrier (BBB) [26] and the disruption of BBB seems to be an essential step in triggering CNS inflammation and subsequent tissue injury [27]. Moreover, astrocytes have been shown to contribute to remyelination as their presence can facilitate the repair of demyelinating lesions by host oligodendrocytes [28,29,30,31].

The last cell type concerns microglia, which are “resting”, characterized by a small cell body with fine, ramified processes. CNS injury triggers a rapid activation of microglial cells and can become phagocytes. They are supposed to potentially influence processing in the CNS, either indirectly via their interactions with astrocytes, or directly buy interacting with synapses [32]. As for astrocytes, microglia/macrophages play a crucial role in both developmental and repairing oligodendrogenesis and myelination. For more details, see review by Traiffort et al., 2020) [33].

### 1.2. Myelination

Myelination is the formation of concentrically laminated membrane structures around the axons; the number of turns and thickness of the layer depends on the axonal diameter [34,35]. In addition to mechanical support to axons, myelin provides metabolic support, regulates repair and plasticity and, most importantly, allows fast conduction of action potential [14]. In the evolutionary tree, myelin first appears in Chondrichthyan Gnathostomes, cartilaginous fish, about 440 million years ago [36,37]. Which later on passed on to higher and more specialized vertebrates with an increased efficiency. Myelination accelerates nerve conduction 20 to 100 times compared to non-myelinated axons of the same diameter, without occupying much space [38,39]. Earlier in 1937, Hodgkin AL, assumed that several internodes contributed to the potential recorded at any points in the axon (for review, see Boullerne AI, 2016) [40]. Sodium channels are located at these intermittent interruptions (nodes of Ranvier), where short portions of the axon are left unwrapped [41].

Myelin contains 40% water and its dry matter consists, unlike most of the body’s plasma membranes, of 70% lipids and 30% proteins [4,42]. In the CNS, cells that synthesize myelin are oligodendrocytes and the major myelin proteins are MBP and PLP, which contribute to the compaction of myelin. Other proteins are found in a smaller percentage, such as CNPase, MAG and MOG [42]. In the murine CNS, MBP and MAG first appear between postnatal day five and seven, followed by MOG. Interestingly, PLP and its isoform DM-20 appear earlier and are therefore part of the early markers of OPCs [6,7,8].

While the protein composition differs substantially in peripheral and central nervous system myelin, lipid species are remarkably similar. The quantitatively most significant lipids in myelin are cholesterol, galactosylceramide and ethanolamine plasmalogen. Different glycolipids such as galactocerebroside, galactosulfatide and ganglioside are also contained in the myelin bilayer [27,43]. In a pioneering study, Kahn and Morell have reported that the majority of brain polyphosphoinositides are related to myelin metabolism [44]. Thus, lipids are essential for the structure of myelin and the change in their composition, during the course of diseases such as MS, leads to the destabilization and the breakdown of myelin [45].

In the CNS, an oligodendrocyte can myelinate up to 40 segments on multiple axons. The process is very well regulated and fast, as much as 5–50 × 10^3^ μm^2^ of membrane a day [46]. The reciprocal communication between axons and oligodendrocytes is very important and its deficiency leads to neurological diseases. For example, in MS, the loss of this communication due to oligodendrocyte death and demyelination leads to a considerable degeneration of axons and astrocytic gliosis [47,48]. MS is a chronic, inflammatory, demyelinating disease of the CNS with secondary axonal damage and loss. The inflammatory process is driven by a T-cell-mediated immune reaction that leads to attacks against both the myelin sheaths and the oligodendrocytes. Demyelination slows down the conduction of electrical impulses along the nerves and leads to the interruption or loss of function. In fact, a large number of broken (transected) axons in active MS lesions have been observed post-mortem. Active lesions are defined as having extensive inflammation which is more pronounced at the earlier stages of MS [49].

### 1.3. Remyelination

The knowledge of this process is of major importance as spontaneous regeneration of myelin (also called “remyelination”) following demyelinating events taking place in the adult CNS. It involves a partial recapitulation of developmental processes [50]. In agreement with this concept, most OPCs generated during the early postnatal period, when the greater part of CNS myelination takes place, differentiate into mature and myelinating oligodendrocytes. However, a subset of these cells remains undifferentiated, scattered throughout all CNS regions, and are recruited in response to demyelinating insults [51]. Rapid remyelination is important to restore metabolic support to the axon, to prevent axon degeneration and subsequent neurological disability, and also to reconstruct the nodes of Ranvier, where are located the voltage-dependent sodium channels necessary for saltatory conduction [52].

Unlike myelination, carried out during development, for which a close association exists between axon diameter and myelin thickness, the thickness of the myelin sheath during remyelination is independent of the diameter of the axon. Moreover, in remyelination, the myelin sheaths are thinner and shorter than the original ones generated during development [53]. Both development and regeneration of myelin, therefore, involve the same key stages involving the proliferation of OPCs and their migration to regions of interest and then their differentiation into mature myelinating oligodendrocytes [50]. However, myelination is mainly driven by axonal signals while remyelination is induced by neuro-inflammation [50].

Animal models of toxin-induced demyelination have shown that the remyelination process is mediated by recruited proliferating OPCs that acquire a mature myelinating phenotype, rather than by myelinating post-mitotic oligodendrocytes remaining present within the lesion [54].

However, whether spared mature oligodendrocytes contribute to remyelination remains a topic of debate. Although, these cells are post-mitotic, non-migratory, and unable to remyelinate upon transplantation into demyelinated lesions [55,56,57].

Recent studies do not exclude a dynamic regeneration of myelin by pre-existing oligodendrocytes in MS. Thus, Yeung et al. utilized the globally elevated content of ^14^C in all biological material synthesized during the peak of atmospheric atomic bomb tests in the early 1960s to determine the age of cellular components [58]. This study suggests that the remyelination of lesions may be produced by pre-existing and not by newly formed oligodendrocytes in MS. This is consistent with electron microscopy studies in animal models, in which surviving mature oligodendrocytes were observed to extend processes and to ensheath demyelinated axons [59,60]. In these studies, authors have observed a depletion of populations of OPCs, and an increased expression of myelin genes in mature oligodendrocytes in MS, which suggests that subsets of mature oligodendrocytes may contribute to remyelination. Therefore, immediate and active therapeutic interventions that inhibit the loss of oligodendrocytes may be crucial to promote remyelination. All these studies bring new important knowledge concerning the remyelination process. However, more in-depth studies and comparisons between species are needed to better define this concept.

## 2. Steroid Hormones and Myelination/Remyelination

### 2.1. Steroid Hormones and Their Nuclear Receptor Signaling Mechanisms

Sex steroids (progestogens, estrogens and androgens) are small molecules with very different properties and biological activities, whose roles go far beyond hormones of reproduction. They are not only produced by the adrenal glands, ovaries and testes and transported to the nervous system [61], but they are also synthesized within the nervous system of both males and females throughout life. For this reason, they have been named neurosteroids. They act via autocrine, paracrine and possibly also intracrine signaling mechanisms [62,63,64]. Therefore, the nervous system is both a source and a target of natural steroids. The term “neuroactive steroid” corresponds to a functional concept and refers to natural hormonal steroids, produced by the nervous system or the peripheral glands, and to synthetic steroids, that modify the activity of neural cells. In the nervous system, neurosteroid synthesis involves successive enzymatic steps, starting with the side-chain cleavage of cholesterol. As the latter cannot cross the blood–brain barrier, it has to be locally produced [65]. The enzyme essential for cholesterol synthesis, 3-hydroxy-3-methylglutaryl-coenzyme A (HMG-CoA) reductase, which converts HMG-CoA into mevalonate, is expressed in the nervous system [66]. Interestingly, the enzyme 5 alpha-reductase, involved in the metabolism of testosterone to its more potent form dihydro-testosterone, is mainly concentrated in the white matter. Thus, high metabolic activity associated with different structures of white matter appears to be directly linked with the formation and maintenance of myelin. Of note, the specific activity of 5 alpha-reductase is elevated in immature brains where myelination is still in process as compared to an adult, fully myelinated brain [67,68]. Similarly, the expression of the steroidogenic enzymes, such as the P450 side chain cleavage (P450scc, CYP11A1), which converts cholesterol into pregnenolone, and the 3-beta-hydroxysteroid dehydrogenase (3β-HSD), which converts pregnenolone to progesterone, are well documented in the CNS [69,70].

Two pathways mediate the biological effects of steroids. The first involves the classical steroid receptors such as progesterone (PR), androgen (AR) and estrogen (ER) receptors, which belong to the nuclear receptor (NR) superfamily and act as transcription factors. A NR, in its inactive form, is bound to heat shock proteins in the cytoplasm. Upon binding of hormone ligand, the NR is released from the heat shock proteins, is phosphorylated, forms a homodimer, and is translocated into the nucleus where it, along with co-regulators, binds to a hormone-response element (HRE), allowing target genes to be transcribed [71] (Figure 1A). These classical steroid receptors are present in different isoforms throughout the nervous system of both sexes [63,69,72,73,74]. The second pathway involves so-called non classical, membrane-associated receptors, mediating rapid and non-genomic steroid actions [75,76,77]. Through their ability to regulate gene expression of many downstream effector genes, steroid hormones and their NRs are involved in various physiological functions, such as development, metabolism, reproduction, behavior and cell homeostasis. Importantly, many steroid receptors have been implicated in neurological, neurodegenerative, and psychiatric diseases, including Alzheimer’s disease, Parkinson’s disease, Huntington’s disease, amyotrophic lateral sclerosis, schizophrenia and MS [70,72,78,79,80,81,82].

### 2.2. Steroid Hormones and Myelin

Steroid hormones have been implicated in demyelinating diseases such as MS, as it is more common in women and as its course differs between women and men [83]. These observations suggest that differences in circulating sex hormones could play a role in its development. The prospective European Pregnancy in Multiple Sclerosis (PRIMS) study, aimed to investigate the influence of pregnancy and delivery on the clinical course of MS, showed that, compared with the pre-pregnancy year, there was a reduction in the relapse rate during pregnancy, most marked during the third trimester, when circulating levels of estrogens and progesterone are highest. There was also a marked increase in the relapse rate during the first three months after delivery, after the drop in sex steroid levels [84].

Importantly, sex-specific neurosteroid synthesis and signaling in the brain may also play a significant role in MS. In men with MS, estrogen synthesis and ERβ signaling are induced, whereas in females with MS, progesterone synthesis and PR signaling are upregulated [85]. A study by Caruso et al. showed increased levels of pregnenolone and decreased levels of 5α-dihydroprogesterone (5α-DHP), 5α-dihydrotestosterone (5α-DHT) and estradiol in the cerebrospinal fluid of adult men with Relapse Remitting (RR) MS [86]. Glial cells in the demyelinating lesions appear to be the primary site of expression of steroidogenic enzymes and steroid receptors [85,87].

However, recent studies have also reported contradictory results concerning the steroid status in MS patients. In fact, levels of important neurosteroids were found to be suppressed in the white matter of MS patients [88] and decreased concentrations of sex steroids were associated with higher serum levels of proinflammatory cytokines [89,90]. On the other hand, in female patients with MS, higher levels of circulating steroids have been measured, including pregnenolone, progesterone, allopregnanolone and their polar conjugates, which may influence neuronal activity and affect the balance between neuroprotection and excitotoxicity [91].

As numerous experimental animal studies have demonstrated an important role of progesterone and testosterone, among other steroid hormones, in the processes of myelination and remyelination (Figure 1B), we focus this review on updated data from our group and others that begin to identify and characterize the roles of these two steroids and their respective NR signaling in myelin development and repair. In particular, recent findings from our laboratory indicate that progesterone and testosterone not only regulate proliferation and differentiation of myelinating cells (Figure 1B), two key events in MS disease, but may also modulate the vulnerability of the myelin sheaths and neurons to toxic insults.

### 2.3. Roles of Progesterone and Progestins in Myelination/Remyelination

As mentioned earlier, progesterone is a neurosteroid that is generated locally in the nervous system, and enzymes required for its synthesis are expressed in both the neurons and glial cells. Progesterone can be converted into 5α-dihydroprogesterone (5α-DHP) by the 5α-reductases and subsequently 5α-DHP can be reduced to 3α,5α-tetrahydroprogesterone (3α,5α-THP) by the enzyme 3α-hydroxysteroid oxidoreductase [63,92,93,94]. Although 3α,5α-THP (allopregnanolone) is a potent positive modulator of the GABA_A_ receptors [95], effects of progesterone are mainly mediated by its direct binding to receptors, including the classic nuclear PRs and their splice variants, the seven transmembrane domain membrane PR (mPR) and the membrane-associated progesterone receptor membrane component 1 (PGRMC1), previously also named 25-Dx. Remarkably, PRs are broadly expressed throughout the brain and can be detected in neural cell type [63,93,96,97]. The nuclear PR is part of the nuclear hormone receptor superfamily of transcription factors. Two forms of PR, a full-length (PR-B, 110 kDa) and an N-terminally truncated form (PR-A, 86 kDa) derived from alternative transcription of the same gene [96,98,99], have been implicated in a variety of functions, including cognition, neuroprotection and dendritic remodeling [96,100].

Different in vitro and in vivo studies showed that progesterone stimulates oligodendrocyte maturation and myelin repair and decreases the severity of disease symptoms in experimental autoimmune encephalomyelitis (EAE), an animal model of MS [19,101,102,103,104,105,106,107,108].

#### 2.3.1. In Vitro Studies

Expression of 3β-HSD by OPCs in cell culture and their capacity to convert progesterone to allopregnanolone only at this particular developmental stage strongly suggest that progesterone synthesis and metabolism may play an important role in oligodendrocyte maturation [101]. Both progesterone and allopregnanolone also modulate oligodendrocyte differentiation [106] and myelin formation [107]. These results were corroborated by our studies using organotypic cerebellar slice cultures, which offer an integrated system for studying developmental events such as myelination [109,110]. Using cerebellar slice cultures from postnatal day seven rats and mice, we demonstrated the role of progesterone in oligodendrocyte maturation and developmental myelination [19,108]. In fact, adding progesterone to the culture medium accelerated the myelination of axons and its effect involved PR, as it was no longer observed in cerebellar slices prepared from PR knockout mice [108]. In addition, in slices from wild-type animals expressing PR, a stimulatory effect of allopregnanolone on myelination involving GABA_A_ receptors could be observed [108]. More importantly, in cerebellar organotypic cultures, a higher density of myelinated axons was observed four days after LPC-induced demyelination in the presence of progesterone, suggesting a remyelinating effect [103]. The 19-norpregnane derivative nestorone, which selectively targets the PR, also efficiently promoted the remyelination of axons by stimulating the proliferation, migration and differentiation of OPCs in cerebellar slice cultures, but at a much lower dose than natural progesterone [103], (see Figure 2). On the contrary, medroxyprogesterone acetate (MPA), another synthetic progestin that binds not only to PR but also to glucocorticoid receptors (GR) and has anti-estrogenic effects, had no effect on myelination or on oligodendrocyte lineage progression [103].

#### 2.3.2. In Vivo Studies

In a model of adult male rat spinal cord injury (SCI) Labombarda and colleagues showed that the administration of progesterone increased the density of OPCs and mature oligodendrocytes and the expression of myelin protein MBP. Furthermore, it decreased the number of astrocytes and microglial cells and down-regulated mRNA expression of inflammatory factors such as interleukin-1β, tumor necrosis factor α, interleukin-6, inducible nitric oxide synthase and cyclooxygenase-2 [111,112,113]. In this model, progesterone also promoted the differentiation of proliferating OPCs into mature oligodendrocytes and exerted myelinating, neuroprotective and anti-inflammatory actions [111,114]. Moreover, progesterone improved neurological outcomes in EAE by enhancing axonal density, decreasing axonal damage and preventing the hyperexpression of the growth associated protein (GAP43). Additionally, EAE mice receiving progesterone showed less inflammatory cell infiltration and recovery of myelin proteins [115,116]. Using different models of demyelination with toxins such as LPC or cuprizone, progesterone has been shown to promote myelin repair and to modulate inflammatory responses [104,117,118]. Our team has also demonstrated that progesterone and nestorone promoted the repair of severe chronic demyelinating lesions induced by feeding cuprizone to female mice for up to 12 weeks. The remyelinating effect of progesterone was PR-dependent, as it was absent in PR knockout mice [104]. These results are strengthened by another report, which indicates that progesterone therapy induces a switch of the microglia phenotype from pro-inflammatory to anti-inflammatory and suppresses inflammasome markers after cuprizone-induced demyelination in mice [119]. Interestingly, treatment with 5a-DHP, a progesterone metabolite, has been shown to increase the expression of MBP in the spinal cord of streptozotocin (STZ)-induced diabetic rats [120].

Even if the cellular mechanisms responsible for the therapeutic effects of progesterone on remyelination are not fully understood, progesterone may participate in remyelination by either stimulating the proliferation and maturation of OPCs, or indirectly by modulating autoimmune and inflammatory processes.

### 2.4. Roles of Androgens in Myelination/Remyelination

#### 2.4.1. Androgen Receptor’s Functions

Androgens, especially testosterone and its metabolite 5α-DHT, play a critical role in sexual development and function. However, they also exert many functions, ranging from the increase in muscle mass and strength, bone remodeling, nervous system development and functions and energy metabolism to general wellbeing. Thus, testosterone depletion results in an increased risk of dysfunction and disease in androgen-responsive tissues, including the brain [121,122,123]. The main effect of testosterone is mediated by the intracellular androgen receptor (AR). The AR gene is located on the X chromosome [124], which allows both males and females to utilize only a single AR allele. In its target cells, testosterone can bind directly to AR or is converted by the 5α-reductase enzymes to 5α-DHT, which has a greater activity at the receptor level and cannot be converted to estrogens. In contrast, testosterone can be metabolized by the aromatase to estradiol (E2) [70,125,126,127]. Mutations in the AR can lead to several diseases in humans such as cancer, Kennedy’s disease (spinal and bulbar muscular atrophy) and complete androgen insensitivity syndrome (CAIS) [128,129,130]. Several mutations of AR have been described. They can result in an incomplete synthesis of the AR, in a loss of its ligand binding capacity or altered interactions with androgen response elements (ARE) on target genes [129]. The discovery of an AR mutation in mice in 1970 by Lyon and Hawkes [131] provided an excellent model for the study of CAIS. Complete androgen insensitivity is referred to as Tfm (for testicular feminization mutation) and affected individuals are males with a feminine appearance, internal and poorly developed testicles. Tfm mice have been used by our team to study the role of AR on myelin (see below).

Previous studies have addressed the cellular distribution of AR in the brain. In the male rat forebrain, AR have been localized mainly in neurons and in some astrocytes, and they were shown to be strongly up-regulated in microglial cells in response to injury [132]. Additionally, the presence of AR has been demonstrated in astrocytes and oligodendrocytes in the prefrontal cortex of adult rhesus monkeys [133]. In the mouse cerebellum, we observed a high expression of AR in Purkinje cells (data not published). Nevertheless, more studies are needed for a comprehensive understanding of the neural cell-type-specific expression and regulation of the AR.

#### 2.4.2. In Vitro and In Vivo Translational Studies and Clinical Trials

An increasing interest exists for AR to join several other promising drug targets for myelin repair as well as for other diseases involving autoimmune and inflammatory destructive mechanisms [122]. In demyelinating disorders such as MS and its experimental models, the neuroprotective effects of androgens have been well documented [31,134,135,136]. Testosterone can exert its effects on target cells via multiple mechanisms and the neural AR was identified as a key target for its remyelinating effects. In fact, in mice fed with 0.2% cuprizone for 12 weeks and in the LPC-lesioned mouse spinal cord, we have shown that testosterone treatment failed to promote myelin repair in Tfm mice with a non-functional AR and, most importantly, after specific conditional invalidation of the AR in neural cells, using Cre-LoxP system (generation and use of AR^NesCre^ mice). Thus, a functional brain AR is required for the remyelination effect of testosterone. Consistently, the potent synthetic AR agonist 7α-methyl-19-nortestosterone (MENT) efficiently stimulated the remyelination of axons [31,134]. Interestingly, MENT is not a substrate for the 5α-reductase enzymes and thus does not stimulate the growth of the prostate, a possible concern with androgen therapy. However, MENT can be converted to biologically active estrogens. This is important, as men taking MENT are not deprived of the benefits of vital estrogen.

Our previous studies have provided evidence that testosterone, its natural metabolites or synthetic derivatives, stimulate the proliferation of OPCs as well as their differentiation into myelinating oligodendrocytes via AR. After the acute demyelination of cerebellar slices in organotypic culture, the remyelinating actions of testosterone could be mimicked by 5α-DHT, a metabolite that is not converted to estrogens, as already mentioned, and blocked by the androgen receptor antagonist flutamide. This may exclude the role of estrogen receptors (ERs) in the remyelinating effects of testosterone [31,134]. All these results highlight the role of testosterone, signaling via the neural AR, in the proliferation of OPCs and their differentiation into myelinating oligodendrocytes (same results as presented in Figure 2).

We have made another interesting and important observation. We have recently shown that the sex differences in oligodendrocyte density and myelin proteins, previously observed by Cerghet et al. [135], are determined by the postnatal increase in testosterone levels and AR expression in male mice [136]. In this study, we reported the unexpected observation that myelin is already sexually differentiated in early postnatal life. The role of AR in determining the sexually dimorphic structure of myelin was further strengthened by using Tfm mice, lacking a functional AR, and by specifically deleting the receptor in neural cells of the CNS of mice (AR^NesCre^ mice). Additionally, we established that sex differences in myelin are already present at postnatal day 10 (P10) and the postnatal treatment by testosterone or 5α-DHT showed their long-lasting masculinizing effects on the density of oligodendrocytes and the structure of the myelin sheaths. In support of these results, we reported, using gas chromatography coupled to tandem mass spectrometry (GC-MS/MS), that brain levels of testosterone and 5α-DHT, both endogenous agonist ligands of the AR, are significantly higher in males when compared with females between postnatal days 0 (P0) and 10 (P10) [136].

In EAE, a sex difference in immune responses is thought to be responsible for the greater susceptibility of females comparatively to males. An implication of testosterone in the susceptibility to autoimmune diseases has indeed been suggested by several studies [137,138]. Testosterone exerts a protective effect in EAE that is mediated at least in part by enhanced production of IL-10 and decreased IL-12 by T lymphocytes. The differences in cytokine production between males and females could be due to a testosterone-induced shift in the immune system toward Th2 immunity. Additionally, castration of male mice has shown deleterious effects on the susceptibility and severity of EAE and treatment with testosterone resulted in the attenuation of symptoms and reduced inflammation [139,140,141]. Recent data also suggest a protective role of testosterone through its action on mast cells by favoring the production of IL-33, a cytokine that promotes a T-helper (Th2) protective response [138]. Moreover, treatment of EAE mice with 5α-DHT resulted in improved clinical scores, and reduced spinal cord gliosis and inflammation [142]. These experimental findings suggest an effective role for androgens as attenuators of the pathogenic Th responses in CNS inflammatory diseases such as MS.

At the clinical level, MS is more common in women even if it presents a faster progression and worse outcomes in men [143]. In accordance, a recent publication described that lower testosterone levels are associated with higher disability in men with MS [144]. Moreover, it could be hypothesized that the later age of onset of MS in men, relative to women, may result from the age-related decline in protective testosterone levels [145]. The first clinical trial by Voskuhl’s team, giving 10 men with relapsing-remitting MS a testosterone gel treatment, showed a shift in peripheral lymphocyte composition by decreasing the percentage of CD4^+^ T cells (TH1 phenotype) and by increasing natural killer (NK) cells [146,147,148]. These exploratory findings suggest that testosterone treatment is safe and well tolerated and has potential protective effects and improved cognition in MS patients. Nevertheless, this is a preliminary study because of the small number of recruited patients [149]. Moreover, larger cohort sizes and multiple routes of administration in addition to transdermal testosterone gels should be explored. Galenic preparations are indeed known to markedly influence the efficacy and safety of steroid treatments [150,151]. As long-term testosterone treatment has been proven safe, and outcomes of a first clinical trial in men with relapsing-remitting MS have been very encouraging, further clinical trials are desirable.

## 3. Conclusions

The nervous system, in addition to be a target for steroid hormones, is the source of a variety of neuroactive steroids, which are synthesized and metabolized by glial cells and neurons. Several studies including ours convincingly show that progesterone and testosterone play a key role in the process of myelination and remyelination. Additionally, their direct neuromodulatory and neuroprotective effects not only maintain the normal physiology of the brain but also lead to a faster recovery in pathology by acting through their specific nuclear receptors. The smaller size of progesterone and testosterone and their natural presence within the circulation, de-novo synthesis within neural tissue and enhanced permeability across the blood–brain barrier point to their high therapeutic potential. The use of their potent synthetic analogs, i.e., nestorone and MENT, with proven efficacy, to minimize the potential undesirable side effects, may offer enhanced benefits for myelin repair in demyelinating diseases. Additionally, understanding the cellular and molecular mechanisms of these steroid hormones in the remyelination processes may contribute to the development of new and safer treatments for MS patients of both sexes.

## Figures and Tables

**Figure 1 ijms-21-03163-f001:**
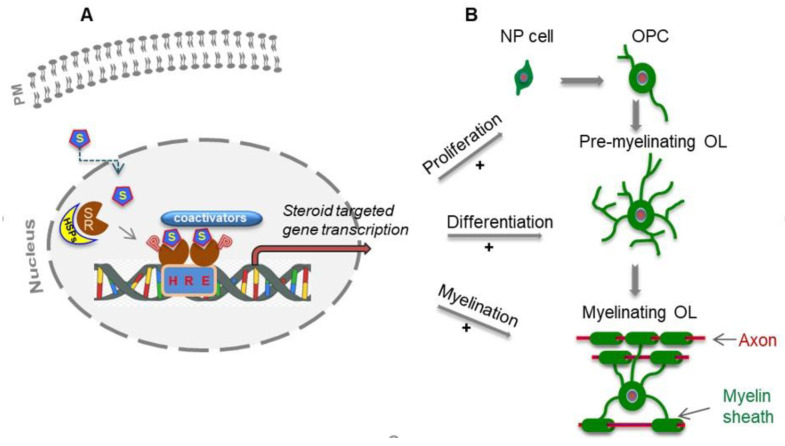
Mechanisms by which progesterone and testosterone act on oligodendrocyte lineage development and myelination. (**A**): Schematic representation of the classical mechanism of actions of steroids (progesterone and testosterone). These neuroactive steroids (S) either derived from the systemic circulation or produced locally in the brain, bind and dissociate nuclear receptors (SR) from HSPs (heat shock proteins). Then, phosphorylated (P) SR translocates to the nucleus and signal through classical transcriptional actions. (**B**): Steroid actions on oligodendrocyte lineage development (proliferation, differentiation and myelination). NP: Neural precursor; OPCs: oligodendrocyte progenitor cells; OL: oligodendrocytes; S: steroids. SR: steroid receptors; HRE: hormone response element; PM: plasma membrane; P: phosphorylation; HSP: heat shock proteins.

**Figure 2 ijms-21-03163-f002:**
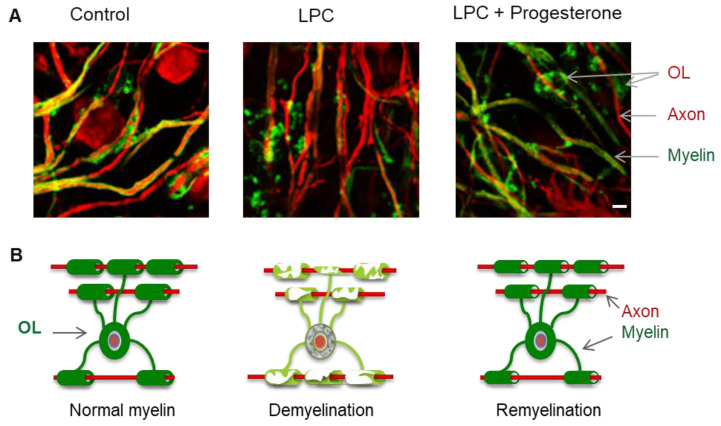
Remyelinating effects of progesterone and nestorone. (**A**): Photos of oligodendrocytes (OL) and myelin in organotypic cultures. Cerebellar slices were taken from 10-day-old mice and maintained in culture for 7 days. Then, slices were demyelinated by lysolecithin (LPC) and treated for 5 days with vehicle (middle panel), progestins (progesterone or nestorone; right panel). Axons of Purkinje cells are labelled with Calbindin marker, in red and myelin is labelled with MBP marker, in green. (**B**): Schematic representation of demyelination and remyelination. Scale bar, 100µM.

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
