# Peer review of "Roles of Progesterone, Testosterone and Their Nuclear Receptors in Central Nervous System Myelination and Remyelination"

_ijms, 2020, doi:10.3390/ijms21093163_

Round 1

Reviewer 1 Report

The authors in the form of a review discuss the stimulatory
effects of progesterone and testosterone on the myelination and
remyelination process. Their summary describes the recent advancement
in the field of myelin biology to treat demyelinating disorders
using natural and synthetic analogues of progesterone and testosterone.
Through figures 1 and 2 they visually summarize the classic mechanism of action of 203 steroids (progesterone and testosterone) and
the remyelinating effects of progesterone and nestorone.
I would like to choose one of the two figures as a possible graphic
extract in order to intrigue and guide the reader in understanding.
The division of labor between in vitro studies and in vitro studies
makes the work easy for the reader to understand.
The conclusions, however, should be somewhat stretched in my opinion.
Finally, the references guide the reader to better understand the state
of the art reported in literature.
It is my opinion that following these small tips, the Review manuscript
entitled Roles of progesterone, testosterone and their 3 nuclear
receptors in central nervous system myelination and 4 remyelination
can be accepted for publication.

Author Response

We were glad to read the positive comments and suggestions of Referee 1.

  • As proposed by the reviewer, figure1 can be chosen as a possible graphic extract
  • Improvements have been done now in the manuscripts and several new references are also added.

Reviewer 2 Report

This manuscript is interesting and brings update on a very worth topic. However, it needs for publishing amendments. There are 61 references added since 2011 previous published work  by the authors.

page 1-2:

The introduction text is quite simplistic and a bit confused about the structure and functions of the 'macroglial cells' or 'macroglia' versus 'microglia'. Each functions of the respective cells (2 types of astrocytes and 2 types of oligodendrocytes) and the microgliocytes, as 'cleaners' but also the barrier cells endothelial and pericytes, seemed forgotten and mixed up in the descriptive and...in the chain of influence.  In addition, even though steroids that diffuse out...how about the astrocyte cascade that influence oligodendrocytes, they also can be interesting parties or not (see Nicaise et al 2019, in this journal). If mentioned, those introductive lines have to make it clear for any unprepared readers who are not in the field. Otherwise, why to make a 'review'?

Lines 31-61: are relying on very recent reviews of reviews...and are not mentioning or forgetting some of the pioneers Nobelists for the cause of myelination [i.e. Eccles, Dale, Loewi, Hodgkin, Huxley) for myelin and saltatory conduction, structure and functions or others (Bunge and Bunge) for the in vitro detection of myelination in CNS and PNS [see Bunge RP, Bunge MB. J Cell Biol 1978; 78: 943-950].

Lines 46-47: Myelin is described as 'lipid bilayer' !?  How about the phospholipids, sphingomyelins, glycolipids'?  Please do not make it so simple! As cited in [5] , proteins are suddenly mentioned... how about completing references and correcting this sentence ?  see also and please refer to:  Morell P 1984. Myelin 2d edition, Plenum Press NY and Raine CS The neuropathology of myelin diseases, also published in: Morell P. 1984. This is not antediluvian but it contains also review of the myelin origin as cited in [4] and more.

In reading many of the published works on the same topic and included in the references, about progesterone and progestins like in those of androgens, there are a large number with modeling/diagrammatic illustrations, would this review suggest that morphology, specially with molecular markers now usable in ultrastructure level, be encouraged to show and see, not just as diagrams or tables the real markers and localizations in cell types the receptors?  Please comment, as an 'image' is better than a long 'discourse'...in the Discussion that was certainly making the data reviewed exciting to find other new clinical potentialities.

Illustration:

Figure 2 should be modified as out of [76]. As viewed, it becomes meaningless because each of the 'Frames' attached are so small to be seen without a microscope. Please convince our reader, otherwise your extracted views, originated from [76], do not need to be repeated here as they are displayed ...

There are references that can be added:

Mellon SH, Vaudry H. Int Rev Neurobiol 2001, 46: 33-78.

Petersen SL et al. J Neurendocrinol 2013; 23: 991-1001.

Author Response

We are very pleased with the positive comments of the referee. We have improved the manuscript by taking into account its suggestions as detailed below.

page 1-2:

The introduction text is quite simplistic and a bit confused about the structure and functions of the 'macroglial cells' or 'macroglia' versus 'microglia'. Each functions of the respective cells (2 types of astrocytes and 2 types of oligodendrocytes) and the microgliocytes, as 'cleaners' but also the barrier cells endothelial and pericytes, seemed forgotten and mixed up in the descriptive and...in the chain of influence.  In addition, even though steroids that diffuse out...how about the astrocyte cascade that influence oligodendrocytes, they also can be interesting parties or not (see Nicaise et al 2019, in this journal). If mentioned, those introductive lines have to make it clear for any unprepared readers who are not in the field. Otherwise, why to make a 'review'?

Answers: We agree with the reviewer, given the very large number of articles and reviews, for more than a century, which have reported studies on the development, structure and functions of these different types of glial cells! In this revised version, the introduction paragraph has been rewritten to present a more balanced view of the glial cells. We now improved the oligodendrocyte part and added a paragraph on the other macro- and micro-glial cells (astrocytes and microglia). We also cited some pioneering studies in this area. Admittedly, this remains exhaustive, but the purpose of this mini-review was above all to shed light on the effects of progesterone and testosterone on myelination and remyelination. Many reviews, some of which are cited in our manuscript, are published by very important and specialist researchers in this field, describe very well these cells.

Lines 31-61: are relying on very recent reviews of reviews...and are not mentioning or forgetting some of the pioneers Nobelists for the cause of myelination [i.e. Eccles, Dale, Loewi, Hodgkin, Huxley) for myelin and saltatory conduction, structure and functions or others (Bunge and Bunge) for the in vitro detection of myelination in CNS and PNS [see Bunge RP, Bunge MB. J Cell Biol 1978; 78: 943-950].

Answers: We apologize to the reviewer! As proposed, we have added some pioneer names in this field such as BUNGE RP, RAINE CS, HUXLEY C, HODGKIN AL and MORLL P. We are sorry if we have not integrated all the proposals of the reviewer, because some works are more oriented towards the SNP than the SNC. Also, we have integrated some reviews presenting the history of myelin (such as Boullerne I, 2016), in which the pioneers in the myelin field are cited.

Lines 46-47: Myelin is described as 'lipid bilayer' !?  How about the phospholipids, sphingomyelins, glycolipids'?  Please do not make it so simple! As cited in [5] , proteins are suddenly mentioned... how about completing references and correcting this sentence ?  see also and please refer to:  Morell P 1984. Myelin 2d edition, Plenum Press NY and Raine CS The neuropathology of myelin diseases, also published in: Morell P. 1984. This is not antediluvian but it contains also review of the myelin origin as cited in [4] and more.

Answers: We admit that this has turned into an oversimplification. We are of course aware of the complexity of the structure and organization of the myelin sheath. We have added information and references on lipids and proteins and have removed misleading sentence “Myelin is described as 'lipid bilayer'.

In reading many of the published works on the same topic and included in the references, about progesterone and progestins like in those of androgens, there are a large number with modeling/diagrammatic illustrations, would this review suggest that morphology, specially with molecular markers now usable in ultrastructure level, be encouraged to show and see, not just as diagrams or tables the real markers and localizations in cell types the receptors?  Please comment, as an 'image' is better than a long 'discourse'...in the Discussion that was certainly making the data reviewed exciting to find other new clinical potentialities.

Answers: the suggestion of the reviewer about the use of steroids and their receptors as molecular markers in ultrastructure is very interesting. However, it is very difficult to propose a state of the art in this area given the difficulty we have with products / marking kits available on the market! I hope, we can do it in the futur!

Illustration:

Figure 2 should be modified as out of [76]. As viewed, it becomes meaningless because each of the 'Frames' attached are so small to be seen without a microscope. Please convince our reader, otherwise your extracted views, originated from [76], do not need to be repeated here as they are displayed ...

Answers: Changes in this figure are now made, allowing to improve the visibility of these figures for the readers.

There are references that can be added:

Mellon SH, Vaudry H. Int Rev Neurobiol 2001, 46: 33-78.

Petersen SL et al. J Neurendocrinol 2013; 23: 991-1001.

Answers: These two references are very interesting. They are now added to the manuscript.

Round 2

Reviewer 2 Report

Congratulations for amendments.

Please spell check 

Line 131  Boulerne A.I.  not I   page 4

Line 671  O'Brien...not O'brien, please

              and the CAPITALIZED (or Latin sized) reference?

The middle frame figure does not show any oligodendrocyte to illustrate similarly the diagrammatic representation. Can that be amended?